# Does Lactate-Guided Threshold Interval Training within a High-Volume Low-Intensity Approach Represent the “Next Step” in the Evolution of Distance Running Training?

**DOI:** 10.3390/ijerph20053782

**Published:** 2023-02-21

**Authors:** Arturo Casado, Carl Foster, Marius Bakken, Leif Inge Tjelta

**Affiliations:** 1Center for Sport Studies, Rey Juan Carlos University, 28933 Madrid, Spain; 2Department of Exercise and Sport Science, University of Wisconsin-LaCrosse, La Crosse, WI 54601, USA; 3Søm Medical Center, 4637 Kristiansand, Norway; 4Departament of Education and Sports Science, University of Stavanger, 4021 Stavanger, Norway

**Keywords:** running, performance, physiological adaptations, endurance sports, lactate, training monitoring

## Abstract

The aim of the present study was to describe a novel training model based on lactate-guided threshold interval training (LGTIT) within a high-volume, low-intensity approach, which characterizes the training pattern in some world-class middle- and long-distance runners and to review the potential physiological mechanisms explaining its effectiveness. This training model consists of performing three to four LGTIT sessions and one VO_2max_ intensity session weekly. In addition, low intensity running is performed up to an overall volume of 150–180 km/week. During LGTIT sessions, the training pace is dictated by a blood lactate concentration target (i.e., internal rather than external training load), typically ranging from 2 to 4.5 mmol·L^−1^, measured every one to three repetitions. That intensity may allow for a more rapid recovery through a lower central and peripheral fatigue between high-intensity sessions compared with that of greater intensities and, therefore, a greater weekly volume of these specific workouts. The interval character of LGTIT allows for the achievement of high absolute training speeds and, thus, maximizing the number of motor units recruited, despite a relatively low metabolic intensity (i.e., threshold zone). This model may increase the mitochondrial proliferation through the optimization of both calcium and adenosine monophosphate activated protein kinase (AMPK) signaling pathways.

## 1. Introduction

On 7 August 2021, 20-year-old Norwegian middle-distance runner Jakob Ingebrigtsen won the 1500 m Olympic title in Tokyo while breaking the Olympic and European records with a time of 3:28.32 (min:s). He also has won the World 5000 m and European 1500 m, 3000 m, 5000 m, and cross-country titles and owns the current indoor 1500 m world record (3:30.60 (min:s)). Further, his brothers Henrik and Filip, also Olympians, won the European 1500 m championships in 2012 and 2016, respectively. Their training pattern was described in a recent article [1] and is considered critical for their development as athletes. While it does not differ greatly from usual training modes in world-class runners [2,3], there is one specific characteristic which makes it unique and innovative: they were typically measuring their blood lactate concentration ([BLa]) during most of their high-intensity training sessions with the intent of matching a specific physiological intensity [1].

The main physiological performance determinants which account for success in distance running events are: maximal oxygen uptake (VO_2max_) [4,5,6]; running economy (RE), defined as steady-state VO_2_ at a given submaximal speed or as the VO_2_ per unit of distance [5,7,8,9]; the ability to sustain a high percentage of VO_2max_ during competition (% VO_2max_) [10,11,12]; the lactate threshold (LT), defined either as the velocity at which a non-linear increase in blood lactate occurs, the maximal lactate steady-state (MLSS), or the velocity corresponding to a blood lactate concentration of 4 mmol·L^−1^ [13]; velocity at LT (vLT)/MLSS [14,15]; and the minimum velocity needed to achieve VO_2max_ (vVO_2max_) [6,16,17]. To improve distance running performance, the training stimulus must enhance one or more of these factors [18].

The training stimulus represents the interaction among training volume (km per week), training frequency, and training intensity designed to enhance the aforementioned performance physiological determinants and performance in distance runners (19). The ideal relationship among these three training variables has, through several decades, been a topic of discussion in both the scientific [19,20,21,22,23,24,25] and coaching [26,27,28,29,30] literature.

However, it remains unclear whether selecting the absolute training intensity composing the training stimulus through the control of an internal training load marker (i.e., blood lactate concentration) to match specific metabolic (relative) intensities may represent a training pattern optimizing the improvement of performance and its physiological determinants in distance runners. Accordingly, the present article aims to describe this training model and its similarities with those considered optimal according to the current scientific literature and examine the potential physiological mechanisms which may support its effectiveness. It would encourage the conduction of further intervention studies testing its influence on performance and its physiological determinants. If this model represents a more efficient training approach than those currently accepted, it may be useful for coaches and athletes, thereby optimizing performance in the latter.

## 2. Historical Trends in Distance Runners’ Training Principles

During the last 100 years, the training principles used by middle- and long-distance runners have been inspired by training theories that provided success for contemporary outstanding runners. To a lesser extent, principles derived from physiological research have contributed to our understanding of how to train runners.

In the 1920s and 1930s, international distance running was dominated by Finnish runners. The Finnish sports professor Lauri Pikhala inspired Pavo Nurmi (nine-time Olympic champion from 1920–1928 in events ranging from 1500 m to 10,000 m and cross country) and other Finnish runners with training principles he brought home from the United States. Their training system during the spring and summer seasons was a precursor to interval training [27]. Nurmi could, for instance, incorporate 6 × 400 m in 60 s into a slow run of 10 to 20 km in the forest [31]. The term “interval training” was introduced in the 1930s by the German coach Woldmar Gerschler and physician Herbert Reindel [27]. Their interval training represented a way to quantify the training load on the basis of repetitive runs to a heart rate of 180 beats/min, with a recovery interval to a heart rate of 120 beats/min. An interval training session consisted of repetitions of shorter (100 m to 400 m) runs. Gerschler was the coach of elite German middle-distance runners, such as Rudolf Harbig, who broke the 800 m world record in 1939 with a time of 1:46.6 (min:s). Importantly, in the 1930s, many years before the advent of portable heart rate monitors, accurately measuring a heart rate of 180 was nearly impossible, and the rationale for choosing “run to 180, recover to 120” is lost to history. Gösta Holmér was the coach of the Swedish runners Gunder Hägg and Arne Anderson who set numerous world records (WR) over distances from 1500 m to 5000 m in the 1940s. Holmér developed “fartlek”, which consisted of intensive efforts of varying distance and duration, interspersed with slower running [32]. It was very similar to Gerschler’s interval training but less formally organized and often conducted “by feel” in the forests rather than on a track. Czech runner Emil Zatopek, multiple-time Olympic champion in events from 5000 m to marathon from 1948 to 1952, typically performed an interval training regime consisting of a very high number of repetitions over 400 m (i.e., 60 × 400 m or 40 × 400 m with a recovery period between repetitions typically of a 200 m jog). The pace used and effort made during these repetitions were submaximal [27].

An interval training regime was also used by Mihály Igloi, who coached Hungarian Sandor Iharos. Iharos broke WRs in the events ranging from 1500 m to 10,000 m during the 1950s. Training intensity during their intervals was higher than that used by Gerschler [27].

In the 1960s, the New Zealand coach Arthur Lydiard criticized the hard interval training regimes, primarily on the grounds that predicting when peak performance would occur was difficult. Lydiard proposed that effective distance running training should be founded on the basis of high volume of continuous low- to moderate-intensity running. He coached his countrymen Peter Snell (three-time Olympic gold medallist in 800 m and 1500 m between 1960 and 1964) and Murray Halberg (5000 m gold in 1960). His training philosophy involved a periodized training pattern. Three main training periods were completed: a 10–12 week preparation period which consisted mainly of high mileage of easy continuous running targeted at reaching 100 miles (160 km) per week, a 6–8 week period characterized by a high volume of hill running, and a 10–12 week competitive period consisting mainly of track interval training at or near race pace leading up to the main competition of the year [26]. In particular, the net effort during the competition period was fairly low, based on Lydiard’s saying, “you can’t train hard and race hard at the same time”. In the same general timeframe, German coach and physician Ernst Van Aaken proposed the Pure Endurance Training Method, which was based on very similar principles as those proposed by Lydiard, but without fixing a specific training volume (i.e., 100 miles per week), using hill repetitions and developing a periodized pattern. Van Aaken coached German runner Harald Norpoth, who achieved a silver medal at the 1964 Olympic Games in the 5000 m event [29].

In the 1970s and 1980s, many athletes who competed at an international level in distance running used a training regime based on Lydiard’s high volume of continuous training principle, but in contrast to Lydiard, they also incorporated sessions of interval training during the preparation period [33,34,35]. The “hard day–easy day” approach to training system is usually attributed to University of Oregon coaches Bill Bowerman and Bill Dellinger (bronze medallist in 5000 m in the 1964 Olympic Games), in which two to three high intensity interval sessions per week were separated by easier days (some with a training volume of <5 km/day) with continuous running [30,36].

From the 1970s and 1980s to the present day, most athletes have used a training regime consisting of two to five weekly sessions of interval training and/or longer tempo runs combined with a relatively high volume of easy and moderate intensity continuous running [33,34,37,38]. A variety of sources have reported that successful distance runners have typically run between 120 and 250 km per week distributed across 11 to 18 sessions [37,38,39,40,41]. Most of these training characteristics have been determined through a ‘trial and error’ approach rather than by the outcomes of intervention studies. Furthermore, apart from Gerschler, internal physiological intensity control during high intensity interval sessions has rarely been proposed as a training strategy to improve performance.

## 3. External and Internal Training Load in Distance Running

The training load, which refers to the interaction between training intensity and training volume, can be understood as either external (i.e., measurable aspects of training occurring externally to the athlete such as volume or intensity (i.e., running speed)) or internal (actual psychophysiological response that the body initiates to cope with the requirements elicited by the external load) [42]. Therefore, external load refers to the actual distance covered and speed achieved during a given training session. In turn, internal load can be measured through the monitoring of heart rate or [BLa]. While external training load represents an important reference to understand the performance evolution during the training process [3], it is generally believed that internal load may be the most accurate indicator of the effort for distance runners [22] as well as for other sports [42]. Accordingly, measuring internal training load (i.e., [BLa]) during training and using that information to control the absolute training intensity (i.e., speed or duration of repetitions) in order to achieve the most optimal stimulus represents a conceptually attractive training protocol which agrees with current recommendations [42].

## 4. Training Volume and Intensity Distribution Analysis in Runners Based on Their Internal Response to Exercise

The aerobic–anaerobic transition as a framework for predicting performance in endurance events was introduced in 1979 by Kindermann et al. [43]. During the last five decades, this framework has been espoused and updated by several scientists using either gas exchange or [BLa] markers [14,40,44,45]. During the last 50–60 years, several definitions related to the LT parameter have been presented [14]. Today it is common to refer to two breakpoints from a plot of the [BLa] during an incremental exercise test in a laboratory. The first threshold (LT1) was named aerobic threshold by Skinner and McLellan [44] and refers to the upper limit of aerobic metabolism. Intensities up to this point could last for hours. The second threshold or second lactate threshold (LT2) that has also been associated with the MLSS is known as the highest constant workload during continuous dynamic work, where there is an equilibrium between lactate production and lactate elimination [14,41,46,47]. At a slightly higher intensity than MLSS, the critical power (CP) concept, which is related to the hyperbolic relationship between speed or power output and the duration for which that speed or power output can be sustained, is an alternative approach to defining the maximal metabolic steady state [48].

According to these concepts, three training intensity zones (see Table 1) for endurance athletes are commonly used [22,49]. Zone 1 represents speeds below first ventilatory threshold or 2 mmol·L^−1^ [BLa]. Zone 2 is represented by speeds between the two ventilatory thresholds or 2 and 4.5 mmol·L^−1^ [BLa] (vLT1 and vLT2, respectively). Zone 3 represents speeds above vLT2 [50]. However, this classification does not differentiate between low- and high-intensity Zone 2 training, nor does it demarcate the different intensity zones that are in Zone 3, such as lactate tolerance and sprint training, being both above the VO_2max_ intensity.

Furthermore, the transition between the different intensity zones does not follow clearly defined limits and are not anchored on exactly defined physiological markers [22]. The relationship between HR and [BLa] will also vary among different runners and in the same athlete across different training periods or seasons [51]. Table 1 describes the type of training performed, typical [BLa], typical % of HRmax, and % VO_2max_ in the various zones for well-trained distance runners. Table 1 uses the intensity scales (i.e., three- and six-zone models) that will be referred to in this article and is elaborated upon according to previous suggestions [1,52,53]. Further mentions of training zones in the present article are referred to by the six-zone scale as z1, z2, …, and z6.

In order to analyze the effect of particular combinations of training volume and intensity in each of these zones, different training intensity distribution models (TID) have been described.

The *pyramidal* model is characterized by a decreasing training volume from z1 to z2 and z3, respectively. Approximately 70–80% of volume is covered in z1, with the remaining 20–30% in z2 and z3 [50].The *polarized* model is characterized by the completion of approximately 80% of the volume at z1, with most of the remaining 20% covered at z3 and as little training as possible in z2 [50].The *threshold* model features a greater proportion of overall volume conducted in z2 (i.e., >35%) than other models.

According to recent reviews [3,54], either polarized or pyramidal approaches improved performance in distance runners to a greater extent than other models, which was also the main conclusion of a recent debate regarding which of the two models was more effective [24,25]. However, a more recent review reported that a pyramidal approach was typically adopted more often in highly trained and elite distance runners, despite the fact that polarized TID also appears to be effective [55]. Most importantly, a high-volume low-intensity approach is carried out in both the pyramidal and polarized TID models.

## 5. Physiological and Performance Development Using Lactate-Guided Threshold Interval Training (LTIT) within a High-Volume Low-Intensity Approach

### 5.1. Physiological Mechanisms Underpinning the Effectiveness of the Use of High Training Volume at Low Intensity

Different hypotheses have been proposed to explain the underpinning mechanisms regarding the reason why a great proportion (70–80%) of overall training volume conducted at low intensity yields optimal performance development in endurance athletes who will race at comparatively high intensities (e.g., low specificity of training). The improvement of endurance performance through high volume of low/moderate continuous training is generated by sustaining increased cardiac output over a prolonged time (therefore augmenting oxygen delivery to working skeletal muscle) and by increased capacity for the oxidative metabolism through mitochondrial biogenesis and capillarization in Type I skeletal muscle fibers [56,57]. Importantly, the mozaic architecture of human skeletal muscle dictates that increased capillarization in Type I skeletal muscle fibers also serves to augment O_2_ delivery in Type II muscle fibers. Two primary signaling pathways for mitochondrial proliferation (both convergent on PGC1-α expression) exist. One is based on calcium signaling, which is more likely used with high-volume training [57,58], and the other is based on signaling derived from adenosine monophosphate (AMP)-activated protein kinase (AMPK) pathway, which is more likely used with high-intensity training, as [ATP] and AMP levels are reduced and increased, respectively [59,60]. As recruiting certain motor units elicited during competitive intensity exercise is needed in order to generate adaptative responses leading to increase mitochondrial density and aerobic metabolism, it can be achieved through the completion of at least a modicum of high-intensity training. The fact that most studies conclude that most of the training volume in distance runners should be covered at easy intensity to optimize performance development implies that adaptive potential of calcium signaling pathway is much larger than that of the AMPK signaling pathway. Accordingly, only relatively small training volume of the latter is needed to reach saturation in the adaptive response using this pathway [58,61].

Alternatively, evidence suggests that some homeostatic disturbances leading to failure to adapt to training (i.e., overtraining or non-functional overreaching) may be related to either inflammatory responses [62] or that slow autonomic recovery following high intensity training [63] may be caused by monotonic loads of high intensity training. These disturbances could lead to a reduction of the capacity for aerobic ATP generation through deficiencies in the mitochondrial electron transport chain or selective delivery of blood flow and/or reductions in maximal cardiac output [24]. Despite the mechanisms involved, quasi-experimental observations [64,65] have suggested the negative effects of an excessive amount of high intensity training. The optimal combination of low- and high-intensity training is typically achieved with a hard day–easy day pattern which avoids monotony during the training process and may act to ensure a sufficient recovery period and to prevent non-functional overreaching. This may augment adaptive responses, such as gene expression for mitochondrial proliferation [50,66]. This specific training pattern is adopted by well-trained and elite long- and middle-distance runners [52,55,64,67,68,69,70]. However, evidence of the exact balance of different types of training on mitochondrial adaptive responses is limited. Particularly in already well-trained athletes, the range of options for achieving additional adaptive responses seems likely to be relatively small. Given the large volume of low-intensity training already performed by high-level athletes, further adaptive responses may largely lie in optimizing adaptive responses in Type II muscle fibers.

### 5.2. Physiological Mechanisms Explaining the Effectiveness of LT2 Intensity Training

It is widely accepted that lactate metabolism serves as a useful index [40,71], although not likely as a cause [72], of muscular fatigue and that a strong correlation exists between lactate accumulation and level of performance in endurance events [15,73,74,75,76]. The relationship between running intensity/speed and [BLa] is widely used to predict and identify performance in distance runners [14,40,74]. A strong correlation between the speed at vLT2/vMLSS and performance in long-distance running has been consistently observed, regardless of the method used to determine these physiological variables [73,77,78,79]. In this sense, Tjeta et al. [51] demonstrated that VO_2max_, RE, and %VO_2max_ explained 89% of the variation in vLT2 among distance runners of national to international level. According to Billat et al. [41], vLT2/vMLSS is a running speed that a well-trained distance runner can sustain for approximately one hour (half-marathon pace for elite runners). Similarly, Roecker et al. [74] found that vLT2/vMLSS was slightly faster than half-marathon pace in 427 competitive runners. This was especially the case for the best runners. As vLT2 during continuous running is close to half-marathon pace, continuous tempo runs from 8–20 km are classified as threshold training in z2 and z3. Tempo runs have been included in the training regime of distance runners from the 1970s up to now [39,70,80,81]. Casado et al. [82] found that elite Kenyan distance runners performed more of their total training volume as tempo runs compared with that in the best Spanish distance runners.

The combination of high training volumes in z1 with moderate volumes in z2 and z3 is a very common pattern in contemporary distance runners. It generated improvements in performance [64,83] or has been associated with very high performance in highly trained and elite middle- and long-distance runners [41,67,68,69]. Furthermore, the use of this approach was reported to be related to either high levels [67,69,70] or an improvement in RE [64,83]. Some research also found either improvements in [64,83,84,85] or were related to high levels of vVO_2max_ [41,69,70]. A few studies, using high volumes in z1 and moderate volumes in z2 and z3, were associated with high levels of VO_2max_ [67,70,86]. Studies using this training pattern also found either improvements in [83,85] or were related with high levels of vLT2 [41,67,69,70]. In any case, there are comparatively few contemporary elite runners who have a total training volume <100 km/week, and most perform >160 km/week [53,55]. This approach has, in most cases, one primary characteristic in common, a high proportion of z2 and z3 training was covered at intensities at or near vLT2 (i.e., high intensity within z2–z3) [41,64,67,68,69,83,87].

The underpinning mechanisms explaining the relationship between training near/at vLT2 and the development of performance and its physiological determinants are not clear. However, it has been hypothesized that the use of this specific exercise intensity improves muscle specific adaptations, including clearing of lactate as opposed to reducing lactate production [88]. Since only recruited motor units are likely to experience increases in mitochondrial number and capillary density, with the exception that increases in capillary density in Type I muscle fibers may benefit O_2_ delivery to Type II muscle fibers, it may be speculated that training near vLT2 optimizes the number of motor units recruited without having to accept the consequences of elevated levels of catecholamines likely to be experienced with z4 training. It is also important to consider that the speed associated with a [BLa] of 4 mmol·L^−1^ is somewhat specific to the pace of competitions in the 10–20 km range, which represents a large percentage of available competitions. Additionally, this velocity can be thought of as «speed work» for marathon runners. Sjodin et al. [76] tried to elucidate the effects of training at the speed associated with onset of [BLa] (vOBLA or speed associated with a [BLa] of 4 mmol·L^−1^) and the mechanisms involved explaining those effects in eight well-trained middle-distance runners. After the addition of one weekly training session consisting of 20 min of continuous running at vOBLA to their usual training regime for 8 weeks, the rate of glycogenolysis during exercise decreased (i.e., reduction of phosphofructokinase/citrate synthase ratio), while the potential to oxidize pyruvate and/or lactate increased (i.e., increased relative activity of heart-specific lactate dehydrogenase). These enzymatic changes were accompanied by an increase in vOBLA and/or a decrease of [BLa] at a same absolute speed. This specific training effect is displayed in Figure 1, which illustrates the evolution/displacement to the right of the lactate/speed curve yielded from an incremental intensity test (see Figure 1).

In addition, these authors [76] found that the runners who were able to maintain [BLa] at 4 mmol·L^−1^ during the 20 min runs experienced greater performance improvement after the training period than runners who allowed [BLa] to “drift”. These data are the first to suggest that relatively tight control of [BLa] during training might be advantageous.

### 5.3. Potential Benefits of Lactate-Guided Threshold Interval Training

In any case, the association between this physiological intensity (i.e., vLT2 or vOBLA) and speed is usually assumed when the run is continuous. However, manipulating the variables composing an interval training session (i.e., repetition velocity, duration, and inter-repetition recovery time) to match vLT2/vMLSS through [BLa] monitoring during the session may allow for the adoption of faster speeds (i.e., faster than those derived from continuous runs) and, thus, optimize the adaptive potential of muscle-fiber-type-specific adaptations required for race pace achievement (i.e., in middle-distance runners). In this sense, Kristensen et al. [89] demonstrated that an interval training program using a higher intensity than that derived from continuous exercise yielded a greater activation of AMP-activated protein kinase in Type II muscle fibers. In this way, conducting training in z2 and z3 while recruiting Type II muscle fibers may provide the mechanical and metabolic advantages both of running close to race pace and at LT2 intensity, respectively. Furthermore, there is an additional advantage of covering interval training at LT2 intensity rather than in z4, which is related to fatigue generation. Burnley et al. [90] found that isometric quadriceps contractions conducted at 10% above the critical torque (i.e., just above LT2 intensity in z4) generated a rate of global and peripheral fatigue four to five times greater than that yielded by the same contractions at 10% below of critical torque (i.e., just below LT2 intensity in z3). These findings agree with the existence of a threshold in fatigue development dependent on whether exercise is carried out at, just below, or just above LT2 intensity. Accordingly, distance runners may benefit from covering some of their interval training sessions at z3 but at faster absolute speeds than vLT2 (assessed through a continuous incremental test) rather than in z4. Nonetheless, this should be done through short duration repetitions so that [BLa] does not progressively rise, as by doing so runners would be able to recover faster from ‘high-intensity’ training sessions. However, the use of intensities within z4–z5 has also been found to be useful in performance development in distance runners (2, 82). A recent systematic review by Rosenblat et al. [91] determined that high-intensity interval training at or below intensities of VO_2max_ allows the improvement in central factors influencing VO_2max_, such as plasma volume, left ventricular mass, maximal stroke volume, and maximal cardiac output. However, peripheral factors influencing VO_2max_, such as skeletal muscle capillary density, maximal citrate synthase activity, and mitochondrial respiratory capacity in Type II fibers can be developed through sprint interval training (i.e., 30 s repetitions) [91]. Therefore, given that these physiological adaptations may not all be achieved through lower intensity training (especially those derived from sprint interval training), a certain but tolerable [65] amount of high intensity training within z4–z6 is also needed to improve performance optimally in distance runners.

## 6. Putting This Training Model into Practice

These theoretical physiological advantages derived from LGTIT within a high-volume low-intensity model are attributed as beneficial by current Norwegian middle- and long-distance runners specialized in events ranging from 1500 m to 10,000 m. In the late 1990s, Marius Bakken (co-author of the present article), a Norwegian elite 5000 m runner, started to test a new training model on himself, which consisted of accumulating a high volume of training at an easy pace, a moderate volume of interval training at threshold intensity while controlling the pace through [BLa] testing during the session and including a low volume of interval training in z5 [92]. He typically covered 180 km overall, conducted four interval training sessions (i.e., two double sessions through a hard day–easy day pattern) at threshold intensity (i.e., at [BLa] ranging from 2 to 4.5 mmol·L^−1^ depending on the specific goal of the session) and one session at z5 per week [92]. Bakken experienced that when following LGTIT, he could perform a much higher training volume compared with that when he carried out interval training in z4. On the assumption that a higher total volume of training is associated with larger adaptive responses, this pattern might be thought of as beneficial. This assumption also agrees with findings of Burnley et al. [90] on the reduced fatigue generation at LT2 intensity when compared with that yielded by z4 training. Bakken developed this training model through a ‘trial and error’ approach and achieved a personal best time in 5000 m of 13:06.39 (min:s), which remains as the second all-time best Nordic best. He transmitted his training knowledge and experience to Gjert Ingebrigtsen, father and former coach of the three Ingebrigtsen brothers, who developed it for the achievement of their well-known athletic performances [92]. Bakken’s approach became a model for contemporary Norwegian runners, and much of the success of Norwegian huge runners at present is based on Bakken̕s training principles. For example, Tokyo 2021 triathlon Olympic champion, Norwegian Kristian Blummenfelt, also used LGTIT [92]. This model has been developed within a successful system of endurance training. Norway, with a population of only 5.5 million, has similar men’s national records in distance running events to those of the USA: 1:42.58, 3:28.32, 7:27.05, and 12:48.45 (min:s) and 2:05:48 (h:min:s) for the 800, 1500, 3000, and 5000m and marathon, respectively. For women, Norwegians have held some of the previous 3000, 5000, and 10,000 m and marathon world records. They also achieved the top national medal count for the cross-country and biathlon skiing events at the 2022 Winter Olympic Games, and both the triathlon 2019 and 2021 World Champion (Gustav Iden) and the aforementioned 2021 Olympic champion (Blummenfelt) are Norwegians [BLa] measurement and scientific testing are/were part of their training processes in most of these athletes.

Interval-training performed with lactate values in z2 and z3 is also classified as threshold training even though the absolute speed at which they are performed can be faster than half-marathon pace. This is especially the case for shorter intervals, and the authors of this article have observed international level distance runners showing 20–25 × 400 m in 64 s average recovering 30 s between repetitions (13:20 (min:s) pace for 5000 m and 26:40 (min:s) for 10,000 m) and 20 × 400 m in 62 s average recovering 60 s between repetitions (12:55 (min:s) pace for 5000 m and, therefore, much faster than half-marathon pace), with [BLa] remaining below 4 mmol·L^−1^. The reason why this can be achieved is that duration of the running time/distance is too short for [BLa] to rise above LT2, and the rest period between repetitions is long enough for [BLa] to return to levels near LT1 but not long enough to decrease under that threshold.

It has been reported that the Ingebrigtsen brothers conducted LGTIT over distances from 2000 m to 3000 m at close to half-marathon pace as well as over distances from 400 m to 1000 m at paces between 5000 m and 10,000 m race paces. The volume of this LGTIT sessions ranges between 8 and 12 km, and the recovery time between repetitions ranges between 20 s and 1.5 min. They often covered two LGTIT sessions in the same day and a fifth specific session at a much higher intensity in z4 or z5 (i.e., 20 × 200 m uphill jogging back in 70 s) (1, 67, 92). Their training intensity has been tightly controlled via measures of heart rate and [BLa] during all interval sessions (1). While the extensive use of LGTIT (i.e., up to four sessions per week) represents a novelty in the training of elite distance runners, several studies have reported the combined use of LT2 and z4/z5 training during the training week. For example, runners may conduct two (or more) different interval training sessions per week covered at LT2 and VO_2max_ intensities, respectively (41, 68–70, 85). On the one hand, the addition of a greater number (i.e., two or three) of ‘high-intensity’ sessions to those typically observed in highly trained and elite runners may represent an advantage in training adaptation, as assimilating this higher training load may provide greater performance improvements. On the other hand, it also may represent an increased risk of injury/overtraining syndrome. Furthermore, the characteristics of LGTIT are different from those accepted in the current literature in distance runners given that traditionally LT2 training is conducted as continuous runs at much slower absolute speeds (31). Furthermore, the use of one sprint training session as well as some strength training sessions have been suggested as part of this training model [92]. In addition, it has been reported that this model involved the completion of a high training volume (i.e., 157–185 km/week) [67,92], which also agrees with the accepted efficacy of high training volume in elite distance runners [2,55,82]. However, the longest run does not exceed 21 km [92]. Finally, while no mention of the periodization approach adopted by these runners through this training model exists, the authors’ personal observations suggest that this training pattern involves the use of a traditional periodization approach, as observed in other elite distance runners [55]. Furthermore, during the competitive period, the z5 hill interval training session should be partly substituted for track workouts targeting competition pace at high [BLa] (i.e., from 5 to 10 mmol·L^−1^), and two LGTIT sessions are removed from the weekly plan. In this way, the goal during the competitive period is to achieve the minimum dose of threshold work which can sustain the previously developed aerobic base allowing for the completion of high volumes of competition pace above z3. This would be consistent with the current literature regarding optimal training periodization in highly trained and elite distance runners and shows a trend from a pyramidal TID during the preparatory period towards a polarized TID during the competitive period [38,55,69,85]. The main goal of the present approach is to improve the speed while keeping [BLa] (and heart rate) stable during LGTIT sessions across the season. An example of speed and physiological responses (i.e., [BLa] and heart rate) responses during three similar LGTIT sessions conducted by Bakken during the 2003–2004 season, leading to his former Nordic 5000 m record of 13:06.39 (min:s) is highlighted in Figure 2 and shows the dramatic fitness improvement derived from the use of the present training model.

Rather than a revolutionary training model, it seems much more the result of an evolutionary pattern, as it is based on training practice which has been developed during the last 100 years of history of training in distance runners. Gerschler trained his athletes within a specific heart rate range; Zatopek covered interval training at submaximal paces and effort; Lydiard and Van Aaken established the need for developing a big aerobic base through high training volumes at an easy pace; and Bowerman demonstrated the usefulness of a hard day–easy day basis. These characteristics were implemented during the training process of the Ingebrigtsen brothers. Other coaches and researchers also assisted in the development of an evidence-based and traditional training pattern, which helped these Norwegian coaches and scientists to generate this new and effective training model for distance runners. An example of one training week in which this training model is being used is described in Table 2.

## 7. Limitations, Future Studies, and Practical Applications

The present article examined the current training regime of some of the best runners in the world and its derived potential physiological benefits on the basis of only observational studies and reports. Therefore, the assumptions stated previously should be taken cautiously since no controlled studies have tested the efficacy of this training model. Furthermore, whereas [BLa] ranges for training zones were suggested according to current recommendations [1,52,53] allowing for interindividual variability, specific values demarcating zones should be detected for each athlete through physiological tests [14]. Additionally, the training characteristics and its effects on performance and its development have been described only in 1500 m and 5000 m runners. Its applicability in other endurance events, such as the marathon, remains uncertain. However, our article presented sufficient evidence showing that these training characteristics display agreement with those reported in the current scientific literature in highly trained and elite distance runners. In addition, their differences may, in fact, be considered advantages of this new training approach from a physiological perspective:The allowance of a greater number of ‘high-intensity’ sessions compared with adopting a usual z4 interval training-based approach.Achieving pre-established goals of internal load during the training session.The possibility of adjusting and individualizing the specific training sessions within the model framework in a periodized approach (i.e., month by month, year by year, etc.). In this way, it is possible to accurately monitor not only the training adaptations being achieved without the need of specific tests but also the response to the different sessions through [BLa] measurements and make individual adjustments to the training program on the basis of this information.Adaptation to altitude training while preventing excessive internal training loads derived from low air’s O_2_ partial pressure, given that [BLa] monitoring ensures that internal load remains at the pre-established levels.

For these reasons, new interventions comparing the physiological and performance effects of the previously described training characteristics with those of traditional training methods in highly trained distance runners are particularly encouraged. In this way, this new training model may represent an evolution of the training characteristics of highly trained and elite distance runners, and if future studies demonstrate its efficacy and safety, it may be implemented in other runners. Training characteristics and intensity distribution characterizing this training model and its derived potential physiological benefits are illustrated in Figure 3.

## Figures and Tables

**Figure 1 ijerph-20-03782-f001:**
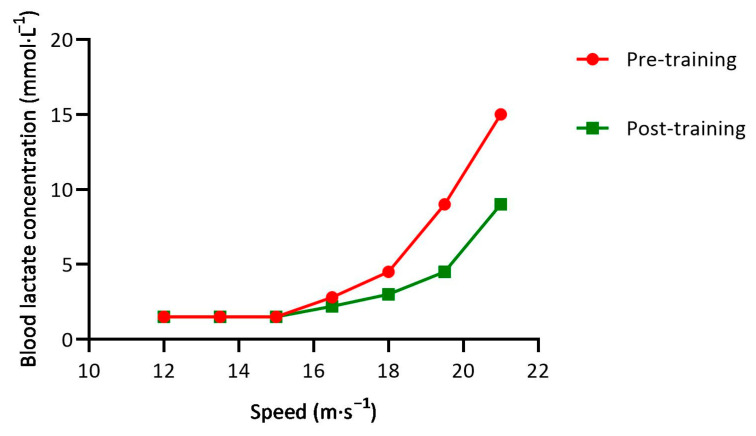
Blood lactate concentration changes between two different incremental intensity tests characterized by a displacement of the lactate/speed curve to the right after including a certain amount of training at the velocity associated with the second lactate threshold during a training period in a hypothetical distance runner.

**Figure 2 ijerph-20-03782-f002:**
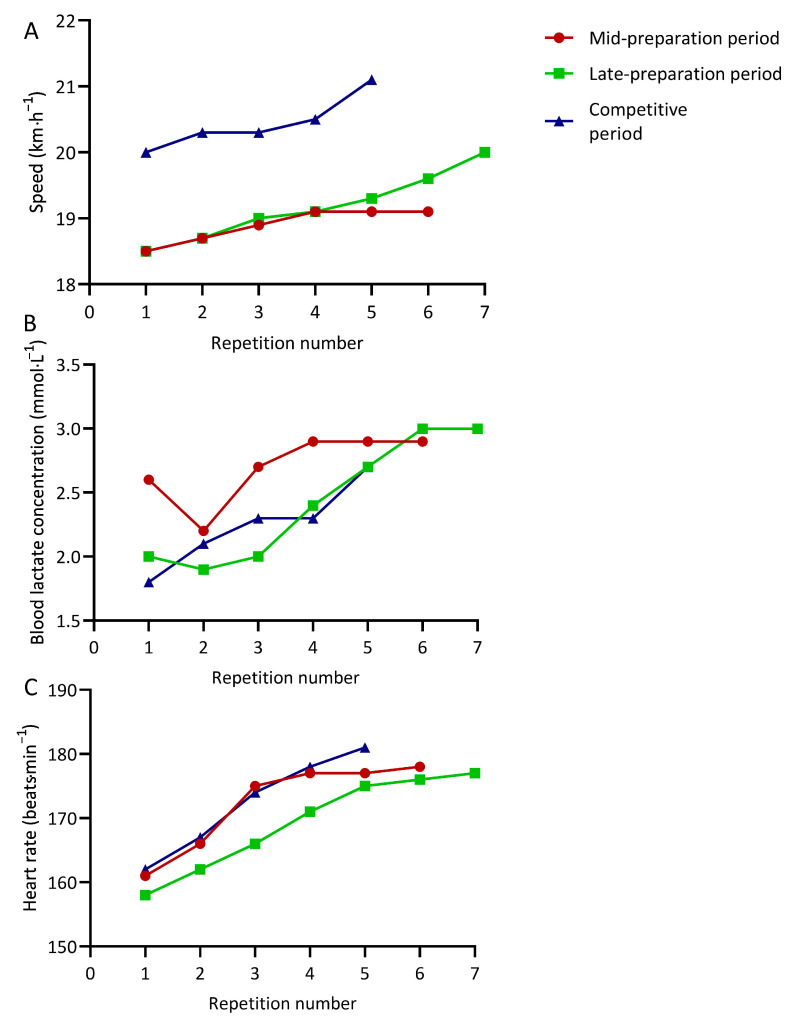
Average speed (**A**) and heart rate (**C**) per repetition, and post-repetition blood lactate concentration (**B**) during three lactate-guided threshold interval training sessions conducted by Marius Bakken across the 2003–2004 season, leading to his former 5000 m Nordic record of 13:06.39 (min:s). Six × 2000, seven × 2000, and five × 2000 m with a recovery time between repetitions of one min were completed in December 2003 (mid-preparation period), February 2004 (late-preparation period), and June 2004 (competitive period), respectively.

**Figure 3 ijerph-20-03782-f003:**
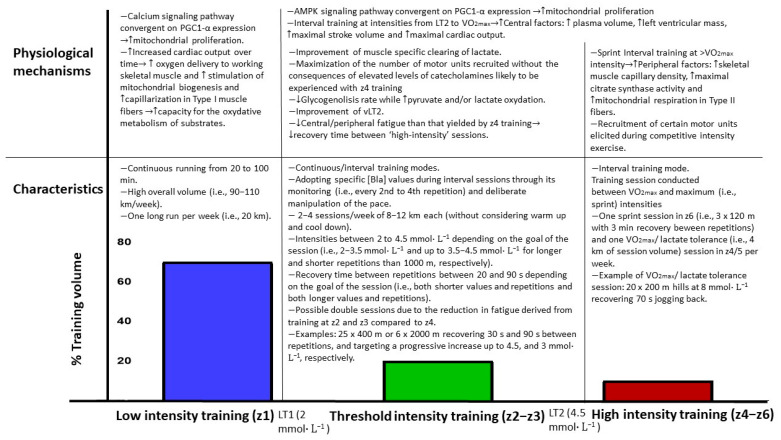
Training characteristics and intensity distribution characterizing the training methodology described in the present article and its derived potential physiological mechanisms leading to performance improvement. LT1: first lactate threshold; LT2: second lactate threshold; vLT2: speed associated to second lactate threshold; VO_2max_: maximum oxygen uptake; vVO_2max_: minimum speed needed to achieve maximum oxygen uptake; z1–6: Zone 1 to Zone 6 according to the 6-zone scale; AMPK: Adenosine monophosphate activated protein kinase; and PGC1-α: Peroxisome proliferator-activated receptor-γ coactivator.

**Table 1 ijerph-20-03782-t001:** Intensity scale for distance runners.

Scale		[BLa]	HR	VO_2max_	RPE	Training Methods
6-Zone	3-Zone	mmol·L^−1^	% Max	%	6–20	
SST (6)	3	n/a	n/a	n/a	n/a	Sprint
VHIT (5)	3	8–18	>97	94–140	18–20	Lactate tolerance (i.e., 800 m and 1500 m pace)
HIT (4)	3	4.5–8	92–97	88–94	16–18	Intensive aerobic interval (i.e., 5000 m pace)
MIT (3)	2	3.5–4.5	87–92	84–88	14–16	Threshold training: interval running (10,000 m pace)
MIT (2)	2	2–3.5	82–87	80–84	12–14	Threshold training: continuous/interval running (marathon pace)
LIT (1)	1	0.7–2	62–82	55–80	9–12	Easy and moderate continuous running

[BLa]: Blood lactate concentration; HR: heart rate; VO_2max_: maximal oxygen uptake; RPE: rate of perceived exertion according to original Borg scale; SST: short sprint training; VHIT; very-high-intensity training, HIT: high-intensity training; MIT: moderate-intensity training; LIT: low-intensity training; n/a: not applicable; numbers in parentheses in the first column refer to each zone of the 6-zone scale and numbers in the second column refer to each zone of the 3-zone scale.

**Table 2 ijerph-20-03782-t002:** Sample training week. Adapted from Bakken [92].

	Morning	Evening
Monday	15 km (z1)	12 km (z1). Sprints (z5) and technique.
Tuesday	5 km (z1). 5 × 6 min at 2.5 mmol·L^−1^ recovering (r.) 1 min between repetitions (z2). 2 km (z1)	5 km (z1). 10 × 1000 m at 3.5 mmol·L^−1^ recovering 1 min between repetitions (z2). 2 km (z1).
Wednesday	16 km (z1). Strength training.	10 km (z1). Sprints (z5) and technique.
Thursday	5 km (z1). 5 × 2 km at 2.5 mmol·L^−1^ recovering 1 min between repetitions (z2). 2 km (z1).	5 km (z1). 25 × 400 m at 3.5 mmol·L^−1^ recovering 30 s between repetitions (z2). 2 km (z1).
Friday	15 km (z1).	Rest.
Saturday	5 km (z1). 20 × 200 m uphill at 8 mmol·L^−1^ recovering 70 s jogging back (z4). 2 km (z1).	10 km (z1).
Sunday	21 km (z1).	Rest.

Z1–5: Zone 1 to Zone 5 according to the 6-zone scale; mmol·L^−1^ is a measure of blood lactate concentration.

## Data Availability

Not applicable.

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
