# Peer review of "Does Lactate-Guided Threshold Interval Training within a High-Volume Low-Intensity Approach Represent the “Next Step” in the Evolution of Distance Running Training?"

_ijerph, 2023, doi:10.3390/ijerph20053782_

Round 1

Reviewer 1 Report

The is a great idea the review the hypothesis of “Lactate-Guided Threshold Interval Training Within a High-Volume Low-Intensity Approach”

You need to clarify in a more specific way about the specific physiological responses in this model.

Author Response

Reviewer 1

R1: The is a great idea the review the hypothesis of “Lactate-Guided Threshold Interval Training Within a High-Volume Low-Intensity Approach”

A: Thanks for the positive comments regarding our manuscript. They contributed to improve its quality.

R1: You need to clarify in a more specific way about the specific physiological responses in this model.

A: Thanks for this comment. Specific physiological responses derived from the use of the present training model are indicated in Figure 3.

Reviewer 2 Report

Dear authors

First of all, thanks for the opportunity to review your work.

The authors aim to describe the lactate-guided threshold interval training as an opportunity to develop better sports achievements in distance running (after to be applied to other activities/modalities that can benefit from it). To achieve this aim, authors made a review, simple, that confronters some of the existent literature regarding this subject and present some very interesting introductions on story and physiological basis of this concept, together with the reflection on this method.

I believe that the work is well written and presents a sound information for the field. I ask why the authors didn´t preform a systematic review? For instance, a systematic review could give more information regarding the affirmation given in page 11 section 7, that the observations are made with the use of only observational studies and reports. There only exists this kind of information? The answer to this question can lead to more changes on the manuscript that, for now, I do not have.

This is my only remark that a systematic review it could be performed increasing the manuscript useability in future studies.

Nevertheless, even with some personal comments in some areas of the manuscript and some ideas that aren’t clearly supported by literature, the reflection made on the manuscript by the authors is very good and I believe it can lead to further discussion on this area of interest.

By other hand, theoretical frameworks are needed to orientate future works. This I believe is achieved with the present work.

Best regards,

Author Response

Reviewer 2

First of all, thanks for the opportunity to review your work.

R2: The authors aim to describe the lactate-guided threshold interval training as an opportunity to develop better sports achievements in distance running (after to be applied to other activities/modalities that can benefit from it). To achieve this aim, authors made a review, simple, that confronters some of the existent literature regarding this subject and present some very interesting introductions on story and physiological basis of this concept, together with the reflection on this method.

A: Thanks for the positive comments regarding our manuscript. They contributed to improve its quality.

R2: I believe that the work is well written and presents a sound information for the field. I ask why the authors didn´t preform a systematic review? For instance, a systematic review could give more information regarding the affirmation given in page 11 section 7, that the observations are made with the use of only observational studies and reports. There only exists this kind of information? The answer to this question can lead to more changes on the manuscript that, for now, I do not have.

A: Thanks for this comment. This is a very important question. We could not perform a systematic review since no research exists in the current literature examining the present training model other than the few observational studies that we already mentioned across the manuscript.

R2: This is my only remark that a systematic review it could be performed increasing the manuscript useability in future studies.

A: Thanks for this comment. Hopefully, the present manuscript may lead to the conduction of randomized control trials which may elucidate to a greater extent the effectiveness of the present training model.

R2: Nevertheless, even with some personal comments in some areas of the manuscript and some ideas that aren’t clearly supported by literature, the reflection made on the manuscript by the authors is very good and I believe it can lead to further discussion on this area of interest.

By other hand, theoretical frameworks are needed to orientate future works. This I believe is achieved with the present work.

A: Many thanks again for all the positive comments already made regarding our work.

Best regards.

Reviewer 3 Report

Manuscript ID: ijerph-2156399

Title: Does Lactate-Guided Threshold Interval Training Within a High-Volume Low-Intensity Approach Represent the “Next Step” in the Evolution of Distance Running Training?

Thank you very much for presenting such an interesting topic. It is undoubtedly very important and I am sure it will be very useful not only for coaches and players but also for experimenters.

This paper is a literature review. The structure of the text and the content, in my opinion, meet the criteria for the preparation and editing of this type of article. The authors have correctly selected the scientific sources and included most of the necessary elements of such a paper:

- justification of the chosen topic;

- what is the history of the research discussed;

- what conclusions can be drawn from the research presented;

- the limitations of the article and the practical application of the research presented.

The section presenting the historical background of the issue is very interesting.

Nevertheless, I comment on the article below:

The Abstract provides an overview of the issue the authors have addressed. However, the reader is not given a clear indication of what the purpose of this paper is and why the authors decided to address such a topic (even though the authors have included this information in Section 7). The content of the Abstract section should be redrafted.

At the end of the Introduction section, there should be a paragraph where the authors should explain why the chosen issue is so important; why the authors chose this particular issue; what benefits such an article can bring and to whom.

In section 4, the authors present experimental data from other authors. On page 4 (3rd paragraph starting "According to these concepts, ....") they describe the zones using shorts: 1 (z1), 2 (z2), 3 (z3). The 6-zone and the 3-zone are listed in Table 1. What happened to zones 4 and 5? Why do the authors use different designations for these zones? Some information about these zones is included in later sections of the paper, but this characterisation makes the content cluttered and the message impossible to understand. A clear system of abbreviations and explanations should be introduced from the beginning.

In Table 1, numbers “6” or “5” etc. appear in the first column next to abbreviations, e.g. SST (6) or VHIT (5) etc... What do they mean? Please explain these if they are necessary. In addition, in the same Table in the second column the digits 1 - 3 are included. What do they mean? Also in the same Table the abbreviation "n/a" appears. What does it mean?

Next to the heading of Table 1, the number of the cited article is missing. Does this mean that it is authored by the authors of this article or, however, is the information borrowed from article No. 50. Please clarify this.

In section 5.2, the authors use the abbreviations 'z2' and 'z3'. Do they mean the same as in previous sections?

The caption to Figures 1, 2 and 3 lacks literature references unless they are original figures by the authors of the paper.

Under Table 2, the authors use the abbreviation Z1-5. Does this mean the same as z1-5? This should be clarified.

In this kind of work, the authors should include a paragraph on what should be done in the next steps in this research. What are their proposals, predictions, expectations....

A paragraph was placed under Figure 3, 'LT1: first lactate threshold; LT2: second lactate threshold; vLT2: speed associated to second lactate threshold; VO2max: maximum oxygen uptake; vVO2max: speed associated to maximum oxygen uptake; z1-6: zone 1 to zone 6; AMPK: AMP activated protein kinase; PGC1-α: Peroxisome proliferator-activated receptor-γ coactivator'.  It is unfinished and I do not understand why it is there. Please explain.

Other comments:

The authors in some places use different expressions for the same content, for example:

- 1,500 or 10.000 or 3000

- 1,500-m or 3,000 m

- mmol.l-1 or mmol·l-1

- [BLa] or [Bla]

- editorial errors: double commas, extra spaces, etc.

In my opinion, the Introduction section does not correctly use the present perfect tense, for example "Further, his brothers, , Henrik and Filip, also Olympians, have won European 1,500-m championships in 2012 and 2016, respectively."

Author Response

Reviewer 3

R3: Thank you very much for presenting such an interesting topic. It is undoubtedly very important and I am sure it will be very useful not only for coaches and players but also for experimenters.

This paper is a literature review. The structure of the text and the content, in my opinion, meet the criteria for the preparation and editing of this type of article. The authors have correctly selected the scientific sources and included most of the necessary elements of such a paper:

- justification of the chosen topic;

- what is the history of the research discussed;

- what conclusions can be drawn from the research presented;

- the limitations of the article and the practical application of the research presented.

The section presenting the historical background of the issue is very interesting.

 A: The authors really appreciate the positive comments made by the reviewer. They contributed to improve its quality.

R3: Nevertheless, I comment on the article below:

The Abstract provides an overview of the issue the authors have addressed. However, the reader is not given a clear indication of what the purpose of this paper is and why the authors decided to address such a topic (even though the authors have included this information in Section 7). The content of the Abstract section should be redrafted.

A: Thanks for this comment. We have now redrafted the abstract section and included the following sentence: “The aim of the present study was to describe a novel training model based on lactate-guided threshold interval training (LGTIT) within a high-volume, low-intensity approach, which characterizes the training pattern in some world-class middle- and long-distance runners, and review the potential physiological mechanisms explaining its effectiveness”.

R3: At the end of the Introduction section, there should be a paragraph where the authors should explain why the chosen issue is so important; why the authors chose this particular issue; what benefits such an article can bring and to whom.

A: Thanks for this comment. The following sentence has been included in the end of the introduction section:

“However, it remains unclear whether selecting the absolute training intensity composing the training stimulus through the control of an internal training load marker (i.e., blood lactate concentration) to match specific metabolic (relative) intensities may represent a training pattern optimizing the improvement of performance and its physiological determinants in distance runners. Accordingly, the present article aims to describe this training model and its similarities with those considered optimal according to current scientific literature and examine the potential physiological mechanisms which may support its effectiveness. It would encourage the conduction of further intervention studies testing its influence on performance and its physiological determinants. If this model represented a more efficient approach of distance running training than those currently accepted, it would help athletes and coaches to improve their performance”.

R3: In section 4, the authors present experimental data from other authors. On page 4 (3rd paragraph starting "According to these concepts, ....") they describe the zones using shorts: 1 (z1), 2 (z2), 3 (z3). The 6-zone and the 3-zone are listed in Table 1. What happened to zones 4 and 5? Why do the authors use different designations for these zones? Some information about these zones is included in later sections of the paper, but this characterisation makes the content cluttered and the message impossible to understand. A clear system of abbreviations and explanations should be introduced from the beginning.

A: Thanks for this comment. That is correct. Therefore, we have now changed this part of the text and not associated z1, z2 and z3 to the zones of the 3-zone scale but rather to those of the 6-zone scale. In this way, we have also included the following sentence in the end of the paragraph being mentioned by the reviewer: “Further mentions to training zones in the present article are referred to the 6-zone scale as z1, z2, …, and z6”.

R3: In Table 1, numbers “6” or “5” etc. appear in the first column next to abbreviations, e.g. SST (6) or VHIT (5) etc... What do they mean? Please explain these if they are necessary. In addition, in the same Table in the second column the digits 1 - 3 are included. What do they mean? Also in the same Table the abbreviation "n/a" appears. What does it mean?

A: Thanks for this comment. We have now included the following sentence in the end of the legend of Table 1: “n/a: not applicable; numbers in parentheses in the first column refer to each zone of the 6-zone scale and numbers in the second column refer to each zone of the 3-zone scale”.

R3: Next to the heading of Table 1, the number of the cited article is missing. Does this mean that it is authored by the authors of this article or, however, is the information borrowed from article No. 50. Please clarify this.

A: Thanks. It means that we have chosen this new value for defining threshold training intensity distribution model.

R3: In section 5.2, the authors use the abbreviations 'z2' and 'z3'. Do they mean the same as in previous sections?

A: Thanks for this comment. It refers to the 6-zone scale as we clarified it just above Table 1 according to the previous reviewer’s comment.

R3: The caption to Figures 1, 2 and 3 lacks literature references unless they are original figures by the authors of the paper.

Thanks for this comment. We made these figures and are entirely original.

R3: Under Table 2, the authors use the abbreviation Z1-5. Does this mean the same as z1-5? This should be clarified.

A: Clarifications were made previously in section 4 so that they refer to 6-zone scale.

R3: In this kind of work, the authors should include a paragraph on what should be done in the next steps in this research. What are their proposals, predictions, expectations....

A: Thanks for this comment. That paragraph was already written prior to Figure 3:

“For these reasons, new interventions comparing the physiological and performance effects of the previously described training characteristics with those of traditional training methods in highly trained distance runners are particularly encouraged. In this way, this new training model may represent an evolution of the training characteristics of highly trained and elite distance runners and if future studies demonstrated its efficacy and safety, it may be implemented in other runners”.

R3: A paragraph was placed under Figure 3, 'LT1: first lactate threshold; LT2: second lactate threshold; vLT2: speed associated to second lactate threshold; VO2max: maximum oxygen uptake; vVO2max: speed associated to maximum oxygen uptake; z1-6: zone 1 to zone 6; AMPK: AMP activated protein kinase; PGC1-α: Peroxisome proliferator-activated receptor-γ coactivator'.  It is unfinished and I do not understand why it is there. Please explain.

A: Thanks. It is the legend of Figure 3. It is separated from the title, which is just above it. We have finished it.

R3: Other comments:

The authors in some places use different expressions for the same content, for example:

- 1,500 or 10.000 or 3000

- 1,500-m or 3,000 m

- mmol.l-1 or mmol·l-1

- [BLa] or [Bla]

- editorial errors: double commas, extra spaces, etc.

A: Thanks for this comment. We have corrected all these mistakes.

R3: In my opinion, the Introduction section does not correctly use the present perfect tense, for example "Further, his brothers, , Henrik and Filip, also Olympians, have won European 1,500-m championships in 2012 and 2016, respectively."

A: Thanks for this comment. We have now corrected that mistake.

Round 2

Reviewer 3 Report

Manuscript ID: ijerph-2156399
Title:  Does Lactate-Guided Threshold Interval Training Within a High-Volume Low-Intensity Approach Represent the “Next Step” in the Evolution of Distance Running Training?

 Thank you very much for taking my comments into account. Thank you for your efforts to improve the text of your work.

However, I still have a doubt about the paragraph under Fig. 3 – “LT1: first lactate threshold; LT2: second lactate threshold; vLT2: speed associated to second lactate threshold; VO2max: maximum oxygen uptake; vVO2max: minimum speed needed to achieve maximum oxygen uptake; z1-6: zone 1 to zone 6; AMPK: Adenosine monophosphate activated protein kinase; PGC1-α: Peroxisome proliferator-activated receptor-γ coactivator.”

The authors wrote in their response:  “…It is separated from the title, which is just above it. We have finished it." However, it still makes no grammatical sense. Since it is written in the same font as the rest of the text it must follow logically from the previous paragraph but it does not. Besides, these are not sentences in the grammatical sense - they do not contain verbs - they are just a list of symbols (and their explanations) placed in the legend of Fig.3. In this form, this paragraph does not fit the text. In my opinion, it should be part of legend under Fig. 3.

Author Response

Thanks for your comment. As we commented previosuly, this paragraph is the legend of figure 3 and is below the title of the figure. In the legends of a figure/table, no verbs should be written, just the explanation of the achronyms existing the figure. According to the reviewer's comments, we have now reduced the size of the font so that it matches with the size of the font of the title of the figure. We also have reduced the space between the title and the legend. However, they can not be in the same line because a title and a legend in a figure are not the same elements and should be separated from each other.